# Hybrid Granularity Distribution Estimation for Few-Shot Learning: Statistics Transfer from Categories and Instances

## Abstract

Distribution estimation (DE) is one of the effective strategies for few-shot learning (FSL). It involves sampling additional training data for novel categories by estimating their distributions employing transferred statistics (*i.e.*, mean and variance) from similar base categories. This strategy enhances data diversity for novel categories and leads to effective performance improvement. However, we argue that relying solely on coarse-grained estimation at category-level fails to generate representative samples due to the discrepancy between the base categories and the novel categories. To pursue representativeness while maintaining the diversity of the generated samples, we propose **H**ybrid **G**ranularity **D**istribution **E**stimation (HGDE), which estimates distributions at both coarse-grained category and fine-grained instance levels. In HGDE, apart from coarse-grained category statistics, we incorporate external fine-grained instance statistics derived from nearest base samples to provide a representative description of novel categories. Then we fuse the statistics from different granularity through a linear interpolation to finally characterize the distribution of novel categories. Empirical studies conducted on four FSL benchmarks demonstrate the effectiveness of HGDE in improving the recognition accuracy of novel categories. Furthermore, HGDE can be applied to enhance the classification performance in other FSL methods. The code is available at: `https://anonymous.4open.science/r/HGDE-2026`

## 1 Introduction

In recent years, deep learning has exhibited remarkable capabilities in computer vision tasks (Rawat & Wang, 2017). These significant advancements heavily rely on the availability of large labeled datasets, which is a time-consuming or even practically infeasible process. To address this challenge, few-shot learning (FSL) has been developed as a solution to recognize novel objects (query set) with only a limited number of labeled samples (support set).

In the typical FSL scenario, a feature extractor like a Convolutional Neural Network (CNN) or Vision Transformer (ViT) is trained on a large set of labeled samples, referred to as the base set. Then a few extracted features from the support set are leveraged to build a classifier for recognizing categories within the query set. Recently, some works (Yang et al., 2021; Liu et al., 2023a;b) have explored the strategy of distribution calibration (DC) (also known as distribution estimation (DE)) to alleviate the data scarcity of the support set. These methods are grounded on the assumption that each feature dimension of the samples follows a Gaussian distribution, and the statistics of the Gaussian distribution can be transferred across similar categories (Salakhutdinov et al., 2012). Consequently, they transfer the statistics of similar base categories to describe the distribution of each novel category in the support set, enabling the sampling of additional training data. As illustrated in Figure 1, given a sample "*Hyena dog*", DC transfers the statistics of the most similar categories (blue ellipse), such as "*Saluki*", "*Tibetan mastiff*" and "*Arctic fox*" to describe the distribution of "*Hyena dog*". Although the previous work Yang et al. (2021) has demonstrated the effectiveness of coarse-grained category-level distribution estimation. We still observe that the prototypes of these categories not only are widely separated in the feature space but also are not sufficiently close to the support category. To validate these observations, we analyze the statistics of similar categories and instances, and the results in Table 5 and Table 6 indicate that similar categories exhibit high diversity

but low similarity, while similar instances exhibit low diversity but high similarity. Consequently, we can infer the success of category-level distribution estimation is due to the provision of diverse information for the novel category, which ensures the diversity of generated samples.

However, due to the discrepancy between the similar base categories and the novel category, these prototypes struggle to accurately represent the novel categories, thus making it challenging to generate representative samples and thus limiting the performance. In contrast, the representative ability of similar instances demonstrated in Table 6 is unpossessed for the similar categories. As depicted in Figure 1, specific samples (located in green dotted ellipse ) within the base categories, such as "*Saluki*", "*Tibetan* mastiff" and "*Arctic fox*", exhibit higher feature similarity to the novel category "*Hyena dog*" in comparison to their category prototypes. Consequently, these samples are more effective in representing the label of "*Hyena dog*".

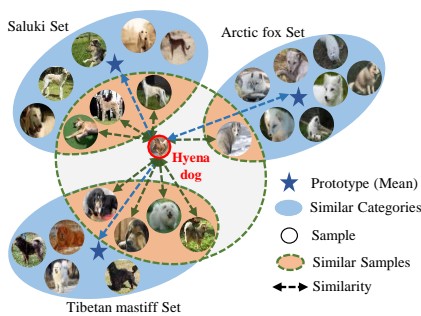

Figure 1: The strategy for category-level and instance-level estimation.

Drawing from the preceding analysis, we conduct the enhancement of the category-level distribution estimation on the fine-grained instance level and propose our **H**ybrid **G**ranularity **D**istribution **E**stimation.

Our HGDE method fuses the statistics estimated from both coarse-grained category-level and fine-grained instance-level data to derive a distribution for each novel category. Specifically, we select several categories and samples in the base set that are similar to the given novel category. We then estimate the mean statistics on the category-level and instance-level for the novel category using a similarity-based refinement. This refined estimation achieved through weighted sum provides a more accurate result compared to the commonly used average operation (e.g., Yang et al. (2021)). Regarding covariance statistics, the category-level estimated covariance is the average of the covariances of the selected categories, and the instance-level estimated covariance is calculated across the selected instances. Additionally, we refine these estimated covariances to filter out less informative components and only preserve principal components using eigendecomposition Yuan & Gan (2017). Finally, we use a simple yet effective linear interpolation to fuse the means and covariances at different levels to obtain the final estimated statistics for the novel category.

The key contributions of our approach can be summarized as follows: (1) We explore the effectiveness of fine-grained instance-level distribution estimations and then transfer statistics from both the coarse-grained category-level and the fine-grained instance-level to ensure the generation of diverse and representative additional samples. (2) We introduce refinement techniques to strengthen the capture of distribution statistics at both levels, respectively. (3) Our HGDE exhibits effectiveness across a wide range of FSL benchmarks and flexibility in its application to various FSL methods.

## 2 RELATED WORKS

### 2.1 TYPICAL METHODS FOR FEW-SHOT LEARNING

Recent FSL works have achieved promising performance by developing a robust feature extractor (backbone) through effective structural design or model training. Meta-learning (Rusu et al., 2019; Lee et al., 2019) is one of the most effective methods due to its ability to quickly adaption. It can be broadly categorized into two types: optimization-based methods and metric-based methods. Optimization-based methods like MAML (Finn et al., 2017) and Reptile (Nichol & Schulman, 2018) focus on learning an initial set of model parameters that can be fine-tuned easily to adapt to new tasks. Metric-based methods, including ProtoNet (Snell et al., 2017) and RENet (Kang et al., 2021), train a neural network to effectively separate samples from the same category, bringing them closer in the feature space, while pushing samples from different categories farther apart.

### 2.2 DATA AUGMENTATION FOR FEW-SHOT LEARNING

Data augmentation is another type of conventional and efficient method to address the issue of limited data and enhance its representational capability. They primarily focus on designing effective

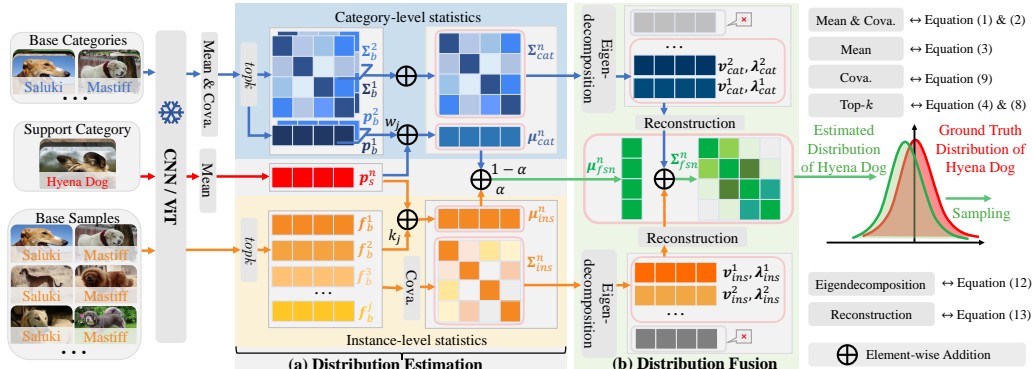

Figure 2: The overview of our approach. We extract features using a frozen CNN or ViT for the given categories and samples. Then we design two-level statistics estimation in **(a) Distribution Estimation** to capture the distribution characteristics. Next, we fuse the two-level statistics to derive the final estimated distribution of the support category in **(b) Distribution Fusion**. Finally, we generate additional samples based on the estimated distribution.

strategies or training auxiliary modules to generate a large number of samples and improve their discriminative ability. Such as Variational Auto-Encoder (VAE) (Schönfeld et al., 2019), Generative Adversarial Network (GAN) (Li et al., 2020). However, due to the time-consuming and expensive nature of training generative networks, another effective augmentation method is Distribution Calibration (DC)(Yang et al., 2021; Li et al., 2022; Liu et al., 2023a), which focuses on estimating the unseen classes by transferring statistics from the seen classes, and then samples are sampled from the distribution with its corresponding label. For example, Yang *et al.* propose LRDC (Yang et al., 2021) which calibrates the novel category distribution based on their similarity with the base classes. Liu *et al.* calibrate the distribution by transferring the statistics from the query samples.

# 3 METHODS

In this section, we first revisit the preliminaries of the FSL. Then we delve into the specifics of our HGDE as depicted in Figure 2. Finally, we describe the classifier training and inference procedures.

## 3.1 PRELIMINARIES

The FSL dataset consists of three main parts: the base set $\mathcal{D}_b$, the support set $\mathcal{D}_s$, and the query set $\mathcal{D}_q$. The base set $\mathcal{D}_b$ contains a substantial number of labeled samples, usually hundreds per category, and is employed to train the feature extractor associated with the label set denoted as $\mathcal{Y}_b$. The support set $\mathcal{D}_s$ and the query set $\mathcal{D}_q$ share the same label space, known as $\mathcal{Y}_{novel}$, which is disjoint from $\mathcal{Y}_b$. The goal of FSL is to accurately classify the unlabeled samples from the query set by leveraging both the base set and the support set. In the support set, $N$ categories are randomly sampled from $\mathcal{Y}_{novel}$, each contributing $K$ samples. This setup is commonly referred to as the $N$-way-$K$-shot recognition problem.

## 3.2 HYBRID GRANULARITY DISTRIBUTION ESTIMATION

Our work operates on features, where all samples are transformed into feature vectors denoted as $\boldsymbol{f} \in \mathbb{R}^d$ using the pre-trained feature extractor, and $d$ is the dimension of the feature vector. Based on the assumption (Yang et al., 2021) that the feature vectors follow a Gaussian distribution, we calculate the prototype (mean) of class $m$ in the base set:

$$\boldsymbol{p}_b^m = \frac{1}{|\mathcal{D}_b^m|} \sum_{i=1}^{|\mathcal{D}_b^m|} \boldsymbol{f}_b^{m,i}, \tag{1}$$

where $\boldsymbol{f}_b^{m,i}$ is the feature of the $i$-th sample from the base class $m$, and $|\mathcal{D}_{\text{base}}^m|$ denotes the total number of samples in the class $m$. Since the feature vector is multidimensional, we use covariance to provide a more comprehensive representation of the variance. The covariance for the features

from base class $m$ is calculated as:

$$\Sigma_b^m = \frac{1}{|\mathcal{D}_b^m| - 1} \sum\nolimits_{i=1}^{|\mathcal{D}_b^m|} (\boldsymbol{f}_b^{m,i} - \boldsymbol{p}_b^m)(\boldsymbol{f}_b^{m,i} - \boldsymbol{p}_b^m)^{\mathrm{T}}. \tag{2}$$

For the support set, we calculate the prototype(s) corresponding to their respective categories:

$$\boldsymbol{p}_s^n = \frac{1}{|\mathcal{D}_s^n|} \sum\nolimits_{i=1}^{|\mathcal{D}_s^n|} \boldsymbol{f}_s^{n,i}, \tag{3}$$

where $\boldsymbol{f}_s^{n,i}$ is the feature of the $i$-th sample from class $n$ in the support set, and $|\mathcal{D}_s^n|$ is the total number of the samples in the class $n$. After the above calculations, we then leverage the statistics of the categories and instances to perform the category-level and instance-level distribution estimations, respectively. Based on these estimated distributions, we fuse them to finally capture the distribution of each support category. The overview of our HGDE is illustrated in Figure 2.

### 3.2.1 CATEGORY-LEVEL DISTRIBUTION ESTIMATION

The statistics of each category in the base set are calculated through Equation 1 and Equation 2, and we transfer these statistics to describe the distribution of categories in the support set. For the given prototype $\boldsymbol{p}_s^n$ from the support set, we select the top $k$ closest base categories:

$$\mathcal{S}_{cat}^n = \{m \mid z_{m,n} \in topk(\mathcal{Z}_{cat}^n)\}, \quad \mathcal{Z}_{\mathrm{cat}}^n = \{-||\boldsymbol{p}_s^n - \boldsymbol{p}_b^m||^2\}_{m=1}^{|\mathcal{Y}_b|}, \tag{4}$$

where $\mathcal{S}_{cat}^n$ stores the set of $k$ nearest base categories corresponding to $\boldsymbol{p}_s^n$, and $\mathcal{Z}_{cat}^n$ collects the Euclidean distance between $\boldsymbol{p}_s^n$ and all base categories. Additionally, $|\mathcal{Y}_b|$ indicates the total number of categories in the base set. Based on the aforementioned observations, we employ the statistics of these similar base categories to estimate the statistics for the category of $\boldsymbol{p}_s^n$:

$$\boldsymbol{\mu}_{cat}^n = \frac{1}{|\mathcal{S}_{cat}^n| + 1} \Big( \sum_{m \in \mathcal{S}_{\mathrm{cat}}^n} w_m \boldsymbol{p}_b^m + \boldsymbol{p}_s^n \Big), \quad \boldsymbol{\Sigma}_{cat}^n = \frac{1}{|\mathcal{S}_{cat}^n|} \sum_{m \in \mathcal{S}_{cat}} \boldsymbol{\Sigma}_b^m. \tag{5}$$

$\boldsymbol{\mu}_{cat}^n$ and $\boldsymbol{\Sigma}_{cat}^n$ are the estimated mean and covariance, respectively. The weights $w_m$ are assigned to the closest base class prototypes to control their contributions in mean estimation:

$$w_m = \min(\frac{1}{\sqrt{-z_{m,n}}} + c_1, 1), \ m \in \mathcal{S}_{cat}^n, \tag{6}$$

where $c_1$ is a constant for decaying $w_j$. For the $N$-way-$K$-shot classification problem, we can derive the $N$ distributions, denoting as the following:

$$\mathcal{U}_{cat} = \{(\boldsymbol{\mu}_{cat}^1, \boldsymbol{\Sigma}_{cat}^1), ..., (\boldsymbol{\mu}_{cat}^n, \boldsymbol{\Sigma}_{cat}^n), ..., (\boldsymbol{\mu}_{cat}^N, \boldsymbol{\Sigma}_{cat}^N)\}. \tag{7}$$

The aforementioned procedure is independent of the value of $K$. Whether it is one shot or more than one shot, we only estimate a single distribution for each category in the support set,

### 3.2.2 INSTANCE-LEVEL DISTRIBUTION ESTIMATION

The instance-level estimation shares similarities with the category-level estimation. We leverage the statistics of the entire feature from the base set, represented as $\boldsymbol{F}_b = [\boldsymbol{f}_b^1, ... \boldsymbol{f}_b^j, ..., \boldsymbol{f}_b^{|\mathcal{D}_b|}]$ to characterize the support set. For the prototype $\boldsymbol{p}_s^n$ from the support set, we select the top $k$ most similar base samples based on their cosine distance:

$$\mathcal{S}_{ins}^n = \{j \mid z_{j,n} \in topk(\mathcal{Z}_{ins}^j)\}, \quad \mathcal{Z}_{ins}^j = \{cos(\boldsymbol{p}_s^n, \boldsymbol{f}_b^j), \ \forall \boldsymbol{f}_b^j \in \boldsymbol{F}_b\}, \tag{8}$$

where $\mathcal{S}_{ins}^n$ represents the associated $k$ nearest base samples, and $\mathcal{Z}_{ins}^j$ is the collections of the distance between $\boldsymbol{p}_s^n$ and all base samples. As the base set contains a large number of samples, there could be a few outliers that influence the selection process. We utilize the cosine distance instead of the Euclidean distance due to its robustness to outliers (Qian et al., 2004). Based on the above calculations, we estimate the instance-level statistics for the category of $\boldsymbol{p}_s^n$:

$$\boldsymbol{\mu}_{ins}^n = \frac{1}{|\mathcal{S}_{ins}^n| + 1} \Big( \sum_{j \in \mathbb{S}_{ins}^n} r_j \boldsymbol{f}_b^j + \boldsymbol{p}_s^n \Big), \quad \boldsymbol{\Sigma}_{ins}^n = \frac{1}{|\mathcal{S}_{ins}^n| - 1} \sum\nolimits_{j=1}^{|\mathcal{S}_{ins}^n|} (\boldsymbol{f}_b^j - \boldsymbol{\mu}_b)(\boldsymbol{f}_b^j - \boldsymbol{\mu}_b)^{\mathrm{T}}, \tag{9}$$

where $\boldsymbol{\mu}_b = 1/|\mathcal{S}_{ins}^n| \sum_{j=1}^{|\mathcal{S}_{ins}^n|} \boldsymbol{f}_b^j$ is the mean of the selected similar base samples, and $r_j$ follows the same idea as in Equation 6, determining the contributions of each selected sample in the estimation:

$$r_j = (z_j)^t + c_2, \ j \in \mathcal{S}_{ins}^n, \tag{10}$$

where $t$ and $c_2$ are both constant to control the range of $r_j$. Depending on the above calculation, we derive the $N$ distributions corresponding to the total categories of the support set:

$$\mathcal{U}_{ins} = \{(\boldsymbol{\mu}_{ins}^1, \boldsymbol{\Sigma}_{ins}^1), ..., (\boldsymbol{\mu}_{ins}^n, \boldsymbol{\Sigma}_{ins}^n), ..., (\boldsymbol{\mu}_{ins}^N, \boldsymbol{\Sigma}_{ins}^N)\}, \tag{11}$$

The Equation 7 and Equation 11 denote the estimated distributions from category-level and instance-level, respectively, and we fuse them to characterize the distribution of categories in the support set.

### 3.2.3 DISTRIBUTION FUSION

The estimated covariance $\boldsymbol{\Sigma}_{cat}^n$ and $\boldsymbol{\Sigma}_{ins}^n$ are based on statistical calculation and can be rewritten as:

$$\boldsymbol{\Sigma}_{cat}^n = \sum_{i=1}^d \lambda_{cat}^i \boldsymbol{v}_{cat}^i {\boldsymbol{v}_{cat}^i}^{\mathrm{T}}, \quad \boldsymbol{\Sigma}_{ins}^n = \sum_{i=1}^d \lambda_{ins}^i \boldsymbol{v}_{ins}^i {\boldsymbol{v}_{ins}^i}^{\mathrm{T}}, \tag{12}$$

where $\lambda_{cat}^i, i = 1, \cdots, d$ and $\lambda_{ins}^i, i = 1, \cdots, d$ are the eigenvalues of $\boldsymbol{\Sigma}_{cat}^n$ and $\boldsymbol{\Sigma}_{ins}^n$, respectively, arranged in descending order. $\boldsymbol{v}_{cat}^i$ and $\boldsymbol{v}_{ins}^i$ are their corresponding eigenvectors. Typically, the feature dimension $d$ is a large number, such as 512 or 640. Based on the covariance matrix analysis in Yuan & Gan (2017), the small values of covariance are less informative and can be regarded as noise components. In order to obtain the statistics more accurately, we reconstruct the estimated covariance to filter out noise components and preserve the principal components as follows:

$$\tilde{\boldsymbol{\Sigma}}_{cat}^n = \sum_{i=1}^L \lambda_{cat}^i \boldsymbol{v}_{cat}^i {\boldsymbol{v}_{cat}^i}^{\mathrm{T}}, \quad \tilde{\boldsymbol{\Sigma}}_{ins}^n = \sum_{i=1}^L \lambda_{ins}^i \boldsymbol{v}_{ins}^i {\boldsymbol{v}_{ins}^i}^{\mathrm{T}}, \tag{13}$$

where $L$ denotes the number of the first largest eigenvalues of $\boldsymbol{\Sigma}_r^{cat}$ and $\boldsymbol{\Sigma}_r^{ins}$. The reconstructed $\tilde{\boldsymbol{\Sigma}}_r^{cat}$ and $\tilde{\boldsymbol{\Sigma}}_r^{ins}$ are more informative and we utilize them to replace the covariance in the $\mathcal{U}_{cat}$ and $\mathcal{U}_{ins}$ respectively. Based on these two-level statistics estimation, we fuse their mean and covariance through a linear interpolation to characterize the distribution for the category of $\boldsymbol{p}_s^n$:

$$\boldsymbol{\mu}_{fsn}^n = \alpha \boldsymbol{\mu}_{cat}^n + (1 - \alpha) \boldsymbol{\mu}_{ins}^n, \quad \boldsymbol{\Sigma}_{fsn}^n = \frac{1}{2} (\tilde{\boldsymbol{\Sigma}}_{cat}^n + \tilde{\boldsymbol{\Sigma}}_{ins}^n) + \boldsymbol{\Delta}. \tag{14}$$

where $\boldsymbol{\Delta}$ is a diagonal matrix with small values added to prevent the singularities (Guerci, 1999), and $\alpha$ is the coefficient to balance the proportion of estimated instance-level and category-level distribution. Finally, we express the distribution of each category in the support set as follows:

$$\mathcal{U}_{fsn} = \{(\boldsymbol{\mu}_{fsn}^1, \boldsymbol{\Sigma}_{fsn}^1), ..., (\boldsymbol{\mu}_{fsn}^n, \boldsymbol{\Sigma}_{fsn}^n), ..., (\boldsymbol{\mu}_{fsn}^N, \boldsymbol{\Sigma}_{fsn}^N)\}. \tag{15}$$

where $\mathcal{U}_{fsn}$ gathers distributions for all categories in the support set. These distributions benefit from both category-level and instance-level statistics, enabling the generation of more diverse and representative samples.

### 3.3 CLASSIFIER TRAINING AND INFERENCE

For each category in the support set, we use the distribution in $\mathcal{U}_{fsn}$ to generate additional samples:

$$\mathcal{D}_{fsn} = \{\boldsymbol{f}_{fsn}^n \mid \boldsymbol{f}_{fsn}^n \sim \mathcal{N}(\boldsymbol{\mu}_{fsn}^n, \boldsymbol{\Sigma}_{fsn}^n), \ \forall (\boldsymbol{\mu}_{fsn}^n, \boldsymbol{\Sigma}_{fsn}^n) \in \mathcal{U}_{fsn}\}, \tag{16}$$

where $\mathcal{D}_{fsn}$ collects the whole generated samples for all categories in the support set. Inspired by the Xu & Le (2022), we assume that the distance between the generated sample and the support prototype reflects its representativeness. To ensure that only the most representative samples are employed for training, we set a threshold $\epsilon$ to constrain samples with distances smaller than $\epsilon$.:

$$\tilde{\mathcal{D}}_{fsn} = \{\boldsymbol{f}_{fsn}^n \mid ||\boldsymbol{f}_{fsn}^n - \boldsymbol{p}_s^n||^2 < \epsilon, \ \forall \boldsymbol{f}_{fsn}^n \in \mathcal{D}_{fsn}\}. \tag{17}$$

The set of all generated samples for each class in the support set, combined with the original samples from the support set, is denoted as $\mathcal{D}_s = \tilde{\mathcal{D}}_{fsn} \cup \mathcal{D}_{\text{support}}$. This combined dataset is then used to train the classifier, which is designed as a fully connected layer optimized with cross-entropy loss:

$$\mathcal{L}_{\text{CE}} = \frac{1}{|\mathcal{D}_s|} \sum_{(\boldsymbol{f}, y) \sim \mathcal{D}_s} \text{CrossEntropy}(\boldsymbol{f}, y), \tag{18}$$

where $y \in \mathcal{Y}_{novel}$ is the label of $\boldsymbol{f}$. In the inference stage, we use the trained classifier to predict the feature from the query set into a specific category directly.

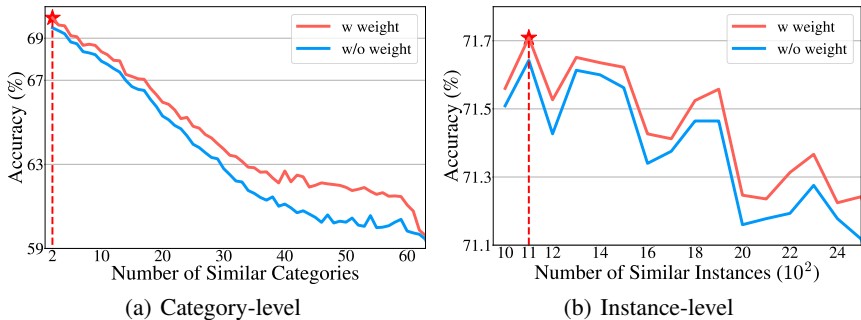

(a) Category-level             (b) Instance-level

Figure 3: The classification accuracy (%) with varying numbers of selection, considering: (a) category-level selection with and without weight described in Equation 6, (b) instance-level selection with and without weight described in Equation 3.2.2.

## 4 EXPERIMENTS

In this section, we first describe the experimental settings, then we perform ablation studies to analyze the contributions of different operations in HGDE. Finally, we compare the performance of our approach with other state-of-the-art (SOTA) methods.

### 4.1 EXPERIMENTAL SETTINGS

**Datasets.** We evaluate our method on four benchmark datasets, *i.e.*, Mini-ImageNet (Vinyals et al., 2016), Tiered-ImageNet (Ren et al., 2018), CUB (Wah et al., 2011), and CIFAR-FS (Krizhevsky et al., 2009). These datasets provide comprehensive evaluation scenarios.
Mini-ImageNet: comprising 100 categories, each containing 600 images, it is split into three segments: 64 base categories for training, 16 novel categories for validation, and the remaining 20 novel categories for testing. Tiered-ImageNet: containing 779165 images from 608 categories, where 351 base categories are used for training, 97 novel categories are used for validation, and the remaining 160 novel categories are used for testing. CUB: including 200 bird categories with 11788 images. We follow the split strategy from Yang et al. (2021), where 100 categories are used for training, 50 for validation, and 50 for testing. CIFAR-FS: derived from CIFAR-100 (Krizhevsky et al., 2009). It is divided into 64 training categories, 16 validation categories, and 20 testing categories, with each category containing 600 images.

**Evaluation.** We report the accuracy results for conducting $N$-way-$K$-shot tasks. In each task, $N$ novel categories are randomly chosen, and within each category, $K$ samples are designated for training. Additionally, 15 samples from each of the $N$ categories are reserved for testing. The reported results are performed on 600 tasks.

**Implementation Details.** We employ the WRN, ResNet, and ViT as feature extractors to ensure a fair comparison with previously published results. All the feature representations are extracted from the feature extractor while keeping it freezing. Our approach involves generating samples alongside support set samples to train the classifier. We employ Adam optimization (Kingma & Ba, 2015) with an initial learning rate of 0.001 and a weight decay of 0.0001. The parameters $c_1$ and $c_2$ are set to 0.3, while the value of $t$ is set to 0.9. Moreover, we generated a total of 1000 samples from the our estimated distribution.

### 4.2 ABLATION STUDY

In the ablation study, we evaluate the effectiveness of various modules of HGDE using the validation set from Mini-ImageNet and the feature extractor employed is the same as Yang et al. (2021).

#### 4.2.1 THE NUMBER OF SELECTING THE BASE CATEGORIES AND SAMPLES.

In this experiment, we investigate the effects of top $k$ and weighted statistics in $\boldsymbol{\mu}_{cat}^n$ and $\boldsymbol{\mu}_{ins}^n$ under $K = 1$, respectively. The category-level selection ranges from 2 to 32, and the instance-level

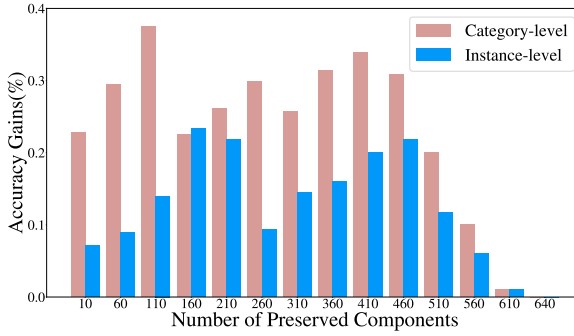

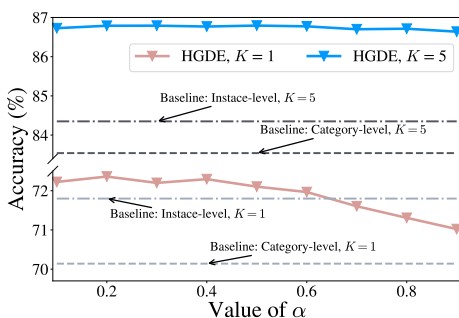

Figure 4: The classification accuracy gains (%) varies with numbers of principal components.

Figure 5: The classification accuracy (%) varies with the generation constraint $\epsilon$.

selection ranges from 1000 to 2500. The results are illustrated in Figure 3. Firstly, we observe that the weighted statistics consistently lead to improved classification results compared to the non-weighted. This improvement is evident across various numbers of selected base categories or base samples. Secondly, when increasing the number of selected base categories, we notice a significant decrease in the classification performance. This decline is attributed to excessive noise introduced by incorporating too many categories. Besides, We also conclude that $k = 2$ for category-level selection and $k = 1000$ for instance-level selection achieves the best performance.

### 4.2.2 THE EFFECTS OF RETAINING PRIMARY COMPONENTS.

In this experiment, we demonstrate the efficacy of reconstruction to determine the optimal principal component retention for estimated covariance. We vary the number of preserved primary components from 10 to 640, aligning with the feature representation dimension, under $K = 1$. The results are shown in Figure 4, where the brown and blue histograms represent the gains achieved for category-level and instance-level estimation, respectively. The results demonstrate the effectiveness of the reconstruction strategy for both category-level and instance-level estimation. For category-level estimation, preserving the first 110 principal eigenvalues and their corresponding eigenvectors yields the best improvements, while retaining the first 160 principal eigenvalues and their corresponding eigenvectors achieves the best improvements for instance-level estimation.

### 4.2.3 THE EFFECTS OF DISTRIBUTION FUSION.

In this study, we assess the impact of the fusing strategy and compare it with the individual category-level and instance-level estimated distribution. Our goal is to determine the optimal fusion parameter $\alpha$ in both $K = 1$ and $K = 5$ settings. The results are illustrated in Figure 5, depicting variations of $\alpha$ between 0.1 and 0.9. Under the $K = 5$, the fused distribution consistently demonstrates superior performance compared to both the estimated category-level and instance-level distributions across the entire range of $\alpha$ values. In the $K = 1$ setting, the fused distribution outperforms the category-level and instance-level distributions with smaller values of $\alpha$. Based on these results, we choose $\alpha = 0.2$ to fuse the estimated category-level and instance-level distributions.

In addition, we conduct a separate analysis of the effects resulting from the estimated category-level and instance-level distributions. The results are summarized in Table 2. which shows that both estimations contribute significantly to the classification performance. The estimated instance-level distribution achieves nearly 2.7% and 1% improvements under $K = 1$ and $K = 5$, respectively. Moreover, combining the generated samples from estimated category-level and instance-level distributions leads to even more remarkable classification gains, as indicated by the second-to-last row in Table 2. Our HGDE, utilizing the fused distribution to generate samples, achieves the most remarkable performance, with over 3.3% and 3% improvements on $K = 1$ and $K = 5$, respectively. These results provide strong validation for the effectiveness of our HGDE.

### 4.2.4 THE EFFECTIVENESS OF GENERATION CONSTRAINT.

In this study, we conduct a comparison analysis of the classification performance with and without a generation constraint strategy in Equation 17. The results are illustrated in Figure 6, where the

Table 1: The classification accuracy (%) with different distributions, where *Cat.* and *Ins.* denote category-level and instance-level estimation, respectively.

| Cat. | Ins. | $K = 1$ | $K = 5$ |
|---|---|---|---|
| ✗ | ✗ | $68.90 \pm 0.83$ | $83.30 \pm 0.53$ |
| ✗ | ✔ | $71.65 \pm 0.56$ | $84.35 \pm 0.56$ |
| ✔ | ✗ | $70.14 \pm 0.55$ | $83.54 \pm 0.59$ |
| ✔ | ✔ | $71.38 \pm 0.88$ | $85.20 \pm 0.55$ |
| **HGDE** | | $\mathbf{72.32} \pm 0.75$ | $\mathbf{86.60} \pm 0.51$ |

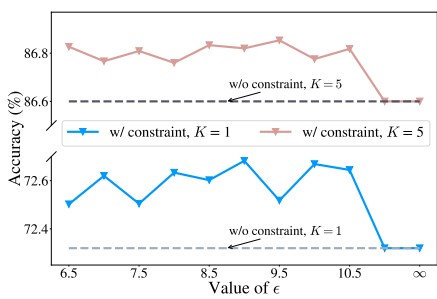

Figure 6: The classification accuracy (%) varies with the fusion ratio $\alpha$.

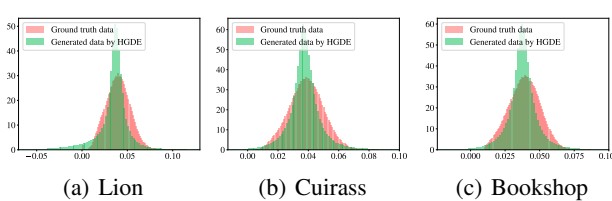

(a) Lion     (b) Cuirass     (c) Bookshop

Figure 7: Feature value distributions.

Table 2: The metric results between generated samples by HGDE and ground truth data. *KL div.*, *mean simi.*, and *var. simi.* denote the KL divergence, mean similarity, and variance similarity, respectively.

| Metric | $K = 1$ | $K = 5$ |
|---|---|---|
| *KL div.* | 0.0325 | 0.0188 |
| *mean simi.* | 0.09925 | 0.9482 |
| *var. simi.* | 0.9800 | 0.9481 |

threshold varies from 6.5 to $\infty$. The findings demonstrate that the generation constraint strategy consistently yields improvements of nearly 0.2% for both the $K = 1$ and $K = 5$ settings. This strategy filters out samples that are more representative and closer to the support set samples, thereby improving the classification performance. However, it's worth noting that as the threshold $\epsilon$ increases beyond a certain point, the improvements begin to taper off. This is because the large threshold fails to adequately filter out the high representative samples. Therefore, in our experiments, we set $\epsilon = 8$.

### 4.2.5 DISTRIBUTION VISUALIZATION.

In this section, we illustrate the sample distribution of our HGDE and ground truth by visualizing the feature value, as depicted in Figure 7. To conduct this analysis, We randomly select several novel categories from the support set, such as "Lion," "Cuirass," and "Bookshop". Then we count the whole feature values of generated data by HGDE and compare them to those of ground truth data for these categories. We find that the value distribution of generated data closely aligns with that of the ground truth data. Besides, we also calculate the metrics, *e.g.*, KL divergence, mean similarity, and variance similarity. The results are shown in Table 2. We observe that the samples generated by our HGDE exhibit low KL divergence and high similarity with ground truth data in the whole dataset perspective. These observations demonstrate the effectiveness of our HGDE approach in addressing the data scarcity problem in FSL.

### 4.2.6 COMPARISONS WITH OTHER METHODS.

We compare the performance of our method with the latest approach with the **Mini-ImageNet** and **Tiered-ImageNet** datasets. Table 3 shows the results, including LEO (Rusu et al., 2019), IFSL (Yue et al., 2020), S2M2 (Mangla et al., 2020), RENet (Kang et al., 2021), FeLMi (Roy et al., 2022), TaFDH (Hu et al., 2023), SUN (Lin et al., 2023), and FewTURE (Lin et al., 2023) At the same time, we apply our approach to recently proposed popular FSL methods, *i.e.*, LRDC, Meta-Baseline, and SMKT. We clearly observe that our approach consistently improves the classification performance across all settings. For different features extracted with various methods, we perform comparable results with the baseline ("LRDC") and obtain the best performance with features from Zhou et al. (2021) ("SMKT + **HGDE**"). In particular, the improvements are generally more significant in the 1-shot setting compared to the 5-shot setting. These findings demonstrate the effectiveness of our approach across different FSL methods and datasets.

Table 3: The accuracies (%) by different methods on the novel categories from Mini-ImageNet Vinyals et al. (2016) and Tiered-ImageNet Ren et al. (2018). † denotes our implementation.

| | Method | Mini-ImageNet | | Tiered-ImageNet | |
|---|---|---|---|---|---|
| | | $K = 1$ | $K = 5$ | $K = 1$ | $K = 5$ |
| WRN-28-10 | LEO Rusu et al. (2019) | $61.76 \pm 0.08$ | $77.59 \pm 0.12$ | $66.33 \pm 0.05$ | $82.06 \pm 0.08$ |
| | IFSL Yue et al. (2020) | $64.12 \pm 0.44$ | $80.97 \pm 0.31$ | $69.96 \pm 0.46$ | $86.19 \pm 0.34$ |
| | S2M2 Mangla et al. (2020) | $64.93 \pm 0.18$ | $83.18 \pm 0.11$ | $73.71 \pm 0.22$ | $88.59 \pm 0.14$ |
| | LRDC Yang et al. (2021) | $68.57 \pm 0.55$ | $82.88 \pm 0.42$ | $74.38^\dagger \pm 0.93$ | $88.12^\dagger \pm 0.59$ |
| | **HGDE** | $\mathbf{69.77} \pm 0.80$ | $\mathbf{84.40} \pm 0.51$ | $\mathbf{74.82} \pm 0.93$ | $\mathbf{89.06} \pm 0.56$ |
| | **HGDE** (Logistic Regression) | $\mathbf{69.85} \pm 0.80$ | $\mathbf{84.63} \pm 0.51$ | $\mathbf{75.22} \pm 0.90$ | $\mathbf{88.60} \pm 0.60$ |
| ResNet-12 | RENet Kang et al. (2021) | $67.60 \pm 0.44$ | $82.58 \pm 0.30$ | $71.61 \pm 0.51$ | $85.28 \pm 0.35$ |
| | FeLMi Roy et al. (2022) | $67.47 \pm 0.78$ | $86.08 \pm 0.44$ | $71.63 \pm 0.89$ | $87.01 \pm 0.55$ |
| | TaFDH Hu et al. (2023) | $67.79 \pm 0.79$ | $82.29 \pm 0.55$ | $72.92 \pm 0.89$ | $85.68 \pm 0.62$ |
| | Meta-Baseline Chen et al. (2021) | $63.17 \pm 0.23$ | $79.26 \pm 0.17$ | $68.62 \pm 0.27$ | $83.74 \pm 0.18$ |
| | Meta-Baseline + **HGDE** | $\mathbf{64.27} \pm 0.78$ | $\mathbf{80.08} \pm 0.57$ | $\mathbf{69.40} \pm 0.96$ | $\mathbf{84.17} \pm 0.60$ |
| ViT | SUN Dong et al. (2022) | $67.80 \pm 0.45$ | $83.25 \pm 0.30$ | $72.99 \pm 0.50$ | $86.74 \pm 0.33$ |
| | FewTURE Hiller et al. (2022) | $72.40 \pm 0.78$ | $86.38 \pm 0.49$ | $76.32 \pm 0.87$ | $89.96 \pm 0.55$ |
| | SMKT$^\dagger$ Lin et al. (2023) | $74.28 \pm 0.18$ | $88.61 \pm 0.48$ | $77.78 \pm 0.94$ | $90.88 \pm 0.54$ |
| | SMKT + **HGDE** | $\mathbf{74.72} \pm 0.84$ | $\mathbf{88.85} \pm 0.47$ | $\mathbf{78.94} \pm 0.89$ | $\mathbf{91.31} \pm 0.54$ |

Table 4: The accuracies (%) by different methods on the novel categories from CUB Wah et al. (2011) and CIFAR-FS Krizhevsky et al. (2009). † denotes our implementation.

| | Method | CUB | | CIFAR-FS | |
|---|---|---|---|---|---|
| | | $K = 1$ | $K = 5$ | $K = 1$ | $K = 5$ |
| WRN-28-10 | S2M2 Mangla et al. (2020) | $80.68 \pm 0.81$ | $90.85 \pm 0.44$ | $74.81 \pm 0.19$ | $87.47 \pm 0.13$ |
| | RENet Kang et al. (2021) | $79.49 \pm 0.44$ | $91.11 \pm 0.24$ | $74.51 \pm 0.46$ | $86.60 \pm 0.32$ |
| | PT-NCM Hu et al. (2021) | $80.57 \pm 0.20$ | $91.12 \pm 0.10$ | $74.64 \pm 0.21$ | $87.64 \pm 0.12$ |
| | H-OT Guo et al. (2022) | $81.23 \pm 0.35$ | $91.45 \pm 0.38$ | $75.40 \pm 0.30$ | $87.50 \pm 0.30$ |
| | LRDC Yang et al. (2021) | $79.56 \pm 0.87$ | $90.67 \pm 0.35$ | $74.98 \pm 0.86$ | $86.54 \pm 0.61$ |
| | **HGDE** | $\mathbf{81.36} \pm 0.78$ | $\mathbf{91.64} \pm 0.41$ | $\mathbf{76.09} \pm 0.85$ | $\mathbf{88.03} \pm 0.59$ |
| | **HGDE** (Logistic Regression) | $\mathbf{81.30} \pm 0.78$ | $\mathbf{91.72} \pm 0.42$ | $\mathbf{76.07} \pm 0.85$ | $\mathbf{88.06} \pm 0.59$ |
| ViT | SUN Dong et al. (2022) | $67.80 \pm 0.45$ | $83.25 \pm 0.30$ | $72.99 \pm 0.50$ | $86.74 \pm 0.33$ |
| | FewTURE Hiller et al. (2022) | $72.40 \pm 0.78$ | $86.38 \pm 0.49$ | $76.32 \pm 0.87$ | $89.96 \pm 0.55$ |
| | SMKT$^\dagger$ Lin et al. (2023) | - | - | $79.68 \pm 0.81$ | $90.64 \pm 0.58$ |
| | SMKT + **HGDE** | - | - | $\mathbf{80.73} \pm 0.84$ | $\mathbf{90.86} \pm 0.56$ |

For **CUB** and **CIFAR-FS** datasets, we employ both WRN-28 and ViT-S as the feature extractors. The compared methods include RENet (Kang et al., 2021), H-OT (Guo et al., 2022), PT-NCM (Hu et al., 2021), S2M2 (Mangla et al., 2020), SUN (Dong et al., 2022), and FewTURE (Hiller et al., 2022). We apply our approach to LRDC and SMKT, and the results are summarized in Table 4. The results demonstrate that our method consistently outperforms the compared methods. Our approach outperforms it significantly by 1.8% and 0.97% under $K = 1$ and $K = 5$ on CUB, respectively. Notably, for the $K = 1$ task on CIFAR-FS, we achieve improvements of over $1\%$ compared to LRDC and SMKT, respectively. In the $K = 5$ task on CIFAR-FS, our approach obtains an accuracy improvement of $1.52\%$ over LRDC. These performance improvements demonstrate the effectiveness of our method in enhancing FSL tasks, even in comparison to robust baselines like SMKT.

## 5 CONCLUSION

In this work, we have proposed Hybrid Granularity Distribution Estimation (HGDE) to address the challenges in few-shot learning. Our approach can be seen as the generalized DC method. Specifically, (1) Our approach ensures diverse and representative additional samples. (2) We introduce statistics refinement to boost the distribution estimation. (3) Our HGDE is simple yet effective in improving the classification performance. Note that the DE-based works including ours all are grounded on the Gaussian distribution followed by features. In the future, we intend to explore alternative assumed distributions, such as non-Gaussian distributions.

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

# A APPENDIX

## A.1 ANALYSIS OF DIVERSITY AND REPRESENTATIVENESS

We assess the diversity and representativeness of category-level and instance-level through mean similarity and multiply distance ("Multi. Dis."), where "Multi. Dis." has the definition as follows:

$$\mathcal{M}_{cat}(i) = \sum_{j,j \neq i} \|\boldsymbol{p}_i - \boldsymbol{p}_j\|^2, \boldsymbol{p}_i, \boldsymbol{p}_j \in \mathcal{S}_{ins},$$

$$\mathcal{M}_{ins}(i) = \sum_{j,j \neq i} \|\boldsymbol{f}_i - \boldsymbol{f}_j\|^2, \boldsymbol{f}_i, \boldsymbol{f}_j \in \mathcal{S}_{ins}, \tag{19}$$

where $\boldsymbol{p}$ denotes the prototype belonging to the similar base category set $\mathcal{S}_{cat}$ and $\boldsymbol{f}$ refers to the feature of the similar instance set $\mathcal{S}_{ins}$. $\mathcal{M}_{cat}$ and $\mathcal{M}_{ins}$ represent the calculated results of "Multi. Dis." in Table 5 and Table 6, respectively. We choose to utilize "Multi. Dis." instead of "variance similarity" because "multiply distance" effectively measures the diversity among similar base categories (samples). In contrast, "variance similarity" essentially shares a similar measurement characteristic to mean similarity. Table 5 presents the category-level statistical results, given the category "Hyena Dog", we observe that as the categories become more similar, the mean similarity increases. Furthermore, each of these similar categories exhibits distances to all other categories that are approximately equivalent. This finding underscores the diversity of the category-level estimation. In contrast, Table 6 showcases the instance-level statistical results. We notice that all similar samples share a high mean similarity. Additionally, their multiply distances are notably smaller in comparison to those presented in Table 5. This finding aligns with the representativeness of the instance-level estimation.

Table 5: The mean similarity between "Hyena Dog" and similar base categories, along with the multiply distance ("Multi. Dis.") for each similar category.

|  | Hyena Dog mean similarity | Multi. Dis. |
|---|---|---|
| Saluki | 0.90 | 1.90 |
| Tibetan mastiff | 0.78 | 1.89 |
| Arctic fox | 0.73 | 1.74 |
| Triceratops | 0.67 | 1.69 |
| Rock Beauty | 0.64 | 1.76 |
| Bolete | 0.62 | 1.82 |

Table 6: The mean similarity between "Hyena Dog" and similar base samples, along with the multiply distance ("Multi. Dis.") for each similar sample.

|  | Hyena Dog mean similarity | Multi. Dis. |
|---|---|---|
| Instance-1 | 0.889 | 1.28 |
| Instance-2 | 0.886 | 1.26 |
| Instance-3 | 0.884 | 1.51 |
| Instance-4 | 0.881 | 1.40 |
| Instance-5 | 0.880 | 1.37 |
| Instance-6 | 0.880 | 1.43 |

## A.2 THE EFFECTIVENESS FOR THE NUMBER OF GENERATED SAMPLES

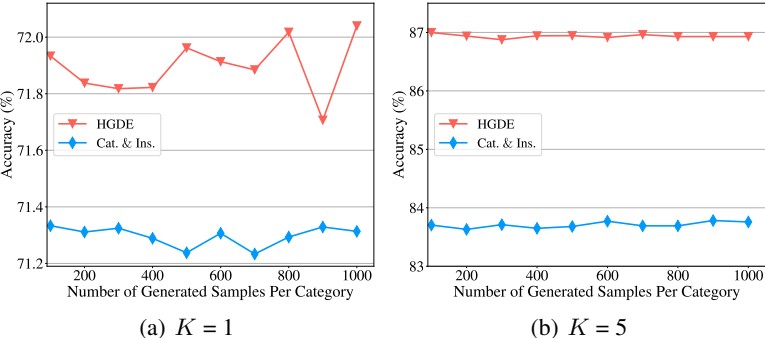

(a) $K = 1$                    (b) $K = 5$

Figure 8: The classification accuracy (%) varies with the number of generated samples

We evaluate the effectiveness of the number of generated samples on the validation set in MiniIm-agenet for both $K = 1$ and $K = 5$, respectively. In Figure 8, we compare the accuracy of samples generated from fused distribution to the total samples $(\mathrm{Cat.\&Ins.})$ generated from the category-level estimated distribution and the instance-level estimated distribution, respectively. It's evident that the samples generated from the fused distribution achieve high accuracy compared to those generated from the estimated category-level and instance-level distributions. This indicates that the fused distribution produces a greater number of high-quality samples beyond the individual category-level and instance-level generated samples.

## A.3 THE RESULTS OF ADDITIONAL SELECTION STRATEGIES.

We conduct experiments on additional selection strategies including random selection in the dataset, all samples in the dataset, and selection from the support class, and the results are shown in Table 7.

Table 7: The classification accuracy (%) with different strategies on Mini-Imagenet.

|  | $K = 1$ | $K = 5$ |
|---|---|---|
| HGDE (random selecting samples in the dataset) | $23.5 \pm 0.59$ | $40.1 \pm 0.77$ |
| HGDE (all samples in the dataset) | $24.4 \pm 0.86$ | $20.32 \pm 0.87$ |
| HGDE (selected samples belong to the same category) | $72.32 \pm 0.33$ | $86.49 \pm 0.25$ |
| HGDE (total calculation) | $69.85 \pm 0.80$ | $86.43 \pm 0.51$ |

Based on the above results, we observe that distribution estimation by randomly selecting samples and selecting all samples in the dataset can't estimate the distribution of the support samples accurately. Consequently, the estimated distribution of these two selection strategies fails to generate valuable samples for few-shot learning. On the other hand, selecting samples that belong to the same category of the support prototypes achieves the best performance, but this strategy entails data leakage, which is practically unfair in the few-shot learning setting.

## A.4 THE ANALYSIS OF TIME CONSUMPTION.

In this section, we evaluate the time consumption of different operations and the total operations of our HGDE in comparison with the LRDC, and the results are summarized in Table 8.

Table 8: The time consumption of different level estimations, LRDC, and our HGDE using Mini-Imagenet on NVIDIA 3090 GPU.

|  | $K = 1$ (600 tasks) | $K = 5$ (600 tasks) |
|---|---|---|
| Category-level Estimation | 272.2s | 273.6s |
| Instance-level Estimation | 303.7s | 305.8s |
| LRDC | 276.3s | 412.6s |
| HGDE (without similarity calculation) | 361.3s | 363.7s |
| HGDE (total calculation) | 366.6s | 368.6s |

The above results demonstrate that our instance estimation requires slightly more time than category estimation in both the 1-shot and 5-shot scenarios. This is due to the larger number of samples compared to the number of categories. However, the time consumption of similarity calculation can

be disregarded in the overall calculation of HGDE. In the comparison between LRDC and HGDE, our HGDE introduces a limited additional training time in the 1-shot scenario. On the other hand, in the 5-shot scenario, HGDE requires less training time because our approach conducts estimation only once for each support prototype, whereas LRDC conducts estimation as many times as the number of support samples. This demonstrates the efficiency of HGDE.

## A.5 THE APPLICATIONS OF HGDE

In this section, we apply our HDGE to the recently popular Vision-Language Models prompt-based methods, *e.g.*, Tip Zhang et al. and APE Zhu et al. (2023), to demonstrate its effectiveness. The results are shown in Table 9.

Table 9: The application of our HGDE to other Vision-Language Model prompt-based methods.

| Method | K=16 | K=8 |
|---|---|---|
| Tip | 62.03 | 61.44 |
| **HGDE**+Tip | 62.42 | 61.76 |
| APE | 63.02 | 62.53 |
| **HGDE**+APE | 63.43 | 63.92 |

The above results show that our HGDE can effectively improve the performance of Tip and APE, which has demonstrated its effectiveness.

