# OpenReview forum: "HYBRID GRANULARITY DISTRIBUTION ESTIMATION FOR FEW-SHOT LEARNING: STATISTICS TRANSFER FROM CATEGORIES AND INSTANCES"
_ICLR.cc/2024/Conference — ICLR 2024 Conference Withdrawn Submission_

### Official Review · Reviewer_3X7E · 2023-10-30

**Soundness:** 3 good
**Presentation:** 3 good
**Contribution:** 2 fair
**Rating:** 6
**Confidence:** 5

**Summary:**

This paper proposes an approach to few-shot learning (FSL) called Hybrid Granularity Distribution Estimation (HGDE), which aims to improve the diversity and representativeness of training data for novel categories. While distribution estimation (DE) has proven effective in FSL by leveraging transferred statistics from similar base categories, it is limited by its coarse-grained estimation at the category level, which may lead to non-representative samples. HGDE addresses this issue by estimating distributions at both category and fine-grained instance levels. It incorporates fine-grained instance statistics from nearest base samples to provide a more representative description of novel categories, ultimately fusing statistics from different granularity levels using linear interpolation. Empirical studies on four FSL benchmarks demonstrate that HGDE significantly enhances the recognition accuracy of novel categories and has the potential to improve classification performance in other FSL methods.

**Strengths:**

1.	One significant advantage of HGDE is its integration of statistics from both coarse-grained category-level and fine-grained instance-level data. This approach ensures that the generated additional samples for novel categories are not only diverse but also representative. This is crucial in FSL, where data availability for novel categories is limited.
2.	The introduction of refinement techniques, such as weighted sum and eigendecomposition, enhances the accuracy of distribution statistics. This refinement process contributes to a more precise estimation of the mean and covariance statistics, which can lead to better model performance.
3.	The paper highlights that HGDE can be applied to various FSL methods, indicating its flexibility and compatibility with existing FSL approaches.

**Weaknesses:**

1.	The proposed method appears to be more complex than some existing FSL approaches. The incorporation of various statistics and refinement techniques may require additional computational resources and might not be suitable for all FSL scenarios, especially those with strict resource constraints.
2.	Apart from the caption below Figure 2, the paper does not provide any additional detailed description for Figure 2.

**Questions:**

see weaknesses

---

> ### Author Response · Authors · 2023-11-20
> **Response to Reviewer(3X7E)**
>
> We appreciate your valuable comments and suggestions for Reviewer(3X7E). We are pleased to respond to your comments and questions as below:
>
> **Q1: The proposed method appears to be more complex than some existing FSL approaches. The incorporation of various statistics and refinement techniques may require additional computational resources and might not be suitable for all FSL scenarios, especially those with strict resource constraints.**
>
>  A1: Thanks for your comments on the complexity of our method. We then explore the additional complexity of our method by comparing the time consumption with simple LRDC, the results are shown below.
>
> |Method|$K$ = 1 (600 tasks) |$K$=5 (600 tasks)|
> |:----|:----|:----|
> |LRDC|276.3s|412.6s|
> |**HGDE**|366.6s|368.6s|
>
> It can be observed that the additional statistics and refinement of our HGDE indeed introduce external time consumption compared to LRDC in the $K=1$ setting. While in the $K=5$ setting, our HGDE makes up for it and achieves better efficiency than LRDC by only conducting estimation for each support prototype other than LRDC which conducts estimation for each support sample. Based on these observations, we imply that the introduced computational overhead is somewhat limited. Our method is performed on the NVIDIA 3090 GPU, and we agree with your opinion that under strict resource constraints, our method may not work well. We will focus on few-shot learning methods under strict resource constraints and consider them as our future work.
>
> **Q2: Apart from the caption below Figure 2, the paper does not provide any additional detailed description for Figure 2.**
>
> A2: Thank you for pointing out this issue, we will provide a detailed description of Figure 2 in the paragraphs of Sec. 3.
> For example: "As illustrated in Figure 2, our HGDE freezes the pre-trained backbone to extract features for both the support set and base set. Then we leverage the statistics of similar categories and instances to estimate the distribution of each novel category, respectively. Based on the estimated distributions, we perform eigenvalue decomposition to eliminate the noise components in the estimated covariances. At last, we fuse these two estimated distributions to obtain the final distribution for the  novel category"

---

> ### Author Response · Authors · 2023-11-23
> **Addressing concerns**
>
> Dear reviewer 3X7E,
>
> Thank you again for your thorough review of our submission. As we approach the end of the author-reviewer discussion phase, we would be grateful if you could inform us whether the responses and revisions paper have addressed your concerns. We would be more than happy to provide further clarifications and revisions if you have any more questions or concerns.

---

### Official Review · Reviewer_Gd5o · 2023-10-30

**Soundness:** 3 good
**Presentation:** 2 fair
**Contribution:** 2 fair
**Rating:** 5
**Confidence:** 4

**Summary:**

This paper argues that previous Distribution estimation (DE) focus on category-level, which is coarse to align the gap between the base categories and the novel categories samples. To fill this gap, this work proposes Hybrid Granularity Distribution Estimation
(HGDE) by leveraging instance-wised information during training, which can lead to more representative description of novel categories. The statistics from different granularity are fused via a linear interpolation. Empirical studies conducted on four FSL benchmarks demonstrate the effectiveness of HGDE

**Strengths:**

- The motivation is clear and novel to me.
- The writing is good and easy to follow.
- The ablation studies are enough and easy to follow.

**Weaknesses:**

- A primary concern arises from the basic hypothesis, that classes with close distant tend to have a similar distribution in the feature space. In my opinion, this is affected by at least two factors, including the optimization of backbone in the pretraining stage and data distribution itself. In the experiments, I appreciate the authors choose to use figures to demonstrate the fidelity of generated features. However, I think it would be better to include some metric-based results on the whole dataset to measure the fidelity of generated features of the proposed method. I think the author needs to design some baselines to show that the proposed method is reasonable, e.g., distribution estimation based all samples in the datasets or random-selected samples.

- The performance gain on recent work SMKT is not significant, why? Also, why the variance of your method is so big?

- What are the extra computation and parameter costs of HGDE over DE?

- Does this method can be applied into recent visual prompt methods?

- I notice that the proposed method has a larger variance compared to the meta-baseline. It would be better to include some clarification.

**Questions:**

See the weakness.

---

> ### Author Response · Authors · 2023-11-20
> **Response to Reviewer(Gd5o)(1/2)**
>
> We appreciate your valuable comments and suggestions of Reviewer(Gd5o). We are pleased to respond to your comments and questions as below:
>
> **Q1: primary concern arises from the basic hypothesis, that classes with close distant tend to have a similar distribution in the feature space. In my opinion, this is affected by at least two factors, including the optimization of backbone in the pretraining stage and data distribution itself. In the experiments, I appreciate the authors choose to use figures to demonstrate the fidelity of generated features. However, I think it would be better to include some metric-based results on the whole dataset to measure the fidelity of generated features of the proposed method. I think the author needs to design some baselines to show that the proposed method is reasonable, e.g., distribution estimation based all samples in the datasets or random-selected samples.**
>
>  **a)** About metric-based results on the whole dataset, we have calculated the KL divergence and cosine distance between our generated samples and the ground truth data to demonstrate the fidelity. The results are summarized below:
>
> ||KL divergence | mean similarity | variance similarity
> |:----:|:----:|:----:|:----:|
> |$K=1$|0.0325|0.9925|0.9482|
> |$K=5$|0.0188|0.9800|0.9481|
>
> It can be observed that samples generated by HGDE demonstrate low KL divergence and high similarity with ground truth data in the whole dataset perspective. This result suggests that generated samples exhibit a close distribution compared with the ground truth data.
>
> **b)** As suggested by the reviewer, we have applied the strategies of all samples and random-selected samples on our HGDE to design baselines, and the results are shown below:
>
>
> ||random-selected samples|all samples|
> |:----:|:----:|:----:|
> |$K=1$|23.5 $\pm$ 0.59|24.4 $\pm$ 0.86|
> |$K=5$|40.1 $\pm$ 0.77|20.32 $\pm$ 0.87|
>
> As demonstrated in the above table, distribution estimation at all samples and random-selected samples can't estimate the distribution of the support samples accurately. Consequently, the estimated distribution of these two selection strategies fails to generate valuable samples for few-shot learning, and we will add these results in the revised manuscript.
>
>  **Q2: The performance gain on recent work SMKT is not significant, why? Also, why the variance of your method is so big?**
>
>  A2: **a)**  We investigate the pipeline of SMKT, and know that SMKT employs two-stage training processing. Our results in the manuscript utilize the backbone after two-stage training. In this training processing, random patch masking is conducted on the training images, resulting in multiple augmented samples for each single image. We imply this strategy acts as a strong data augmentation which plays a similar role as our HGDE. To validate our implication, we conduct experiments based on the backbone after first-stage training and make a comparison with the official first-stage results of SMKT as below:
>
> |Method|$K=1$|$K=5$|
> |:----:|:----:|:----:|
> |SMKT(self-supervised)|60.93|80.38|
> |HGDE(self-supervised)|63.58|81.91|
>
> We observe that using the first-stage backbone, our HGDE outperforms SMKT by 2.45\% and 1.53\% under $K=1$ and $K=5$, respectively.
>
> **b)** We appreciate the thorough review of performance variation, and we analyze the gap in the performance variation arising from the difference in the number of tasks. The variation of SMKT is conducted on 2000 tasks, while the results of our HGDE are only conducted on 600 tasks as described in Section 4. To validate this, we conduct an additional experiment where HGDE is evaluated on 2000 tasks for a fair comparison. The result is shown below:
>
> |Method|$K=1$|$K=5$|
> |:----|:----|:----|
> |SMKT|74.28 $\pm$ 0.18|88.61 $\pm$ 0.48|
> |**HGDE**(600 tasks)|**74.74 $\pm$ 0.84**|**88.87 $\pm$ 0.47**|
> |**HGDE**(2000 tasks)|**74.73 $\pm$ 0.31**|**88.85 $\pm$ 0.36**|
>
>
> It can be observed that the variation of HGDE decreases as the number of tasks increases to 2000 and becomes comparable to SMKT.
>
>  **Q3: What are the extra computation and parameter costs of HGDE over DE?**
>
> A3: Our HGDE does not introduce any learnable parameters, as well as DE. As for the extra computation, we measure it as the extra time consumption under the same hardware condition. As shown in the tab below, compared with the classical DE method, i.e. LRDC, and our implemented DE method (category/instance-level), our HGDE introduces limited time consumption. It is worth noting that, compared with LRDC under $K=5$, our HGDE demonstrates better efficiency because our HGDE only conducts estimation once for each support prototype while LRDC conducts estimation as many times as the number of support samples.
>
> ||$K$ = 1 (600 tasks) |$K$=5 (600 tasks)|
> |:----:|:----:|:----:|
> |LRDC|276.3s|412.6s|
> |DE(only categoty-level)|272.2s|273.6s|
> |DE(only instance-level)|303.7s|305.8s|
> |**HGDE**|366.6s|368.6s|

---

> ### Author Response · Authors · 2023-11-20
> **Response to Reviewer(Gd5o)(2/2)**
>
> **Q4: Does this method can be applied into recent visual prompt methods?**
>
> A4: To the best of our knowledge, pure visual prompt methods receive little attention in the few-shot learning tasks. However, we investigate the adaption of vision-language models in few-shot learning, in which, prompt techniques are utilized, such as Tip[r1] and APE[r2]. To expansively validate the effectiveness of our HGDE, we integrate HGDE into Tip[r1] and APE[r2] and achieve performance improvement as shown below.
>
> |Method|$K=16$|$K=8$|
> |:----|:----|:----|
> |Tip|62.03|61.44|
> |**HGDE+Tip**|**62.42**|**61.76**|<br>
> |APE|63.02|62.53|
> |**HGDE+APE**|**63.43**|**62.92**|<br>
>
> [r1] Tip-adapter: Training-free adaption of CLIP for few-shot classification. ECCV 2022
> [r2] Not all features matter: Enhancing few-shot CLIP with adaptive prior refinement. ICCV 2023
>
> **Q5: I notice that the proposed method has a larger variance compared to the meta-baseline. It would be better to include some clarification.**
>
> A5: We explore the pipeline of the Meta-baseline and locate two reasons for its smaller variation: first, it applies a consistent sampling strategy which decreases the randomness of the evaluation compared to our HGDE which uses random sampling; second, Meta-baseline also employs 800 testing tasks, while our HGDE only employs 600 testing tasks.
>
>  We further increase the number of testing tasks of our HGDE to 800 and employ the consistent sampling strategy as the Meta-baseline. The results are shown below.
>
> |Method|$K=1$|$K=5$|
> |:----|:----|:----|
> |Meta-Baseline|63.17 $\pm$ 0.23|79.26 $\pm$ 0.17|
> |**HGDE**(600 tasks)|**64.27 $\pm$ 0.78** | **80.08 $\pm$ 0.57**|
> |**HGDE**(800 tasks)|**64.29 $\pm$ 0.26** |**80.07 $\pm$ 0.23** |
>
> Based on the results, we find that our HGDE achieves a comparable variation as the Meta-baseline under the same evaluation settings.

---

> ### Author Response · Authors · 2023-11-23
> **Addressing concerns**
>
> Dear reviewer Gd5o,
>
> Thank you again for your thorough review of our submission. As we approach the end of the author-reviewer discussion phase, we would be grateful if you could inform us whether the responses and revisions paper have addressed your concerns. We would be more than happy to provide further clarifications and revisions if you have any more questions or concerns.

---

### Official Review · Reviewer_dMbi · 2023-11-02

**Soundness:** 2 fair
**Presentation:** 2 fair
**Contribution:** 2 fair
**Rating:** 5
**Confidence:** 5

**Summary:**

This paper proposed Hybrid Granularity Distribution Estimation (HGDE), which estimates distributions at both coarse-grained category and fine-grained instance levels. Apart from coarse-grained category statistics, the proposed method incorporates external fine-grained instance statistics derived from nearest base samples to provide a representative description of novel categories. Then the proposed method fuses the statistics from different granularity through a linear interpolation to finally characterize the distribution of novel categories.

**Strengths:**

1. The illustrations are clearly presented, colorful, and easy to understand.

2. This paper is well-written and easy to read.

**Weaknesses:**

1. Motivation is not described clearly and insightfully. Why utilize fine-grained instances to estimate distributions?

2. Utilizing HGDE to estimate distributions is simple, and increases the amount of computation but the gains are limited compared with counterparts. Also, the category estimation has been explored in the community, and this degrades the contribution and novelty of this paper.

3. In section 3.2.2, the top k most similar base samples are selected based on cosine distance, if the selected samples belong to the same class as the support prototypes, is it still reasonable for learning of the distribution, or even counterproductive?

4. What are the pre-trained datasets of the chosen feature-extracting networks? Also, the increased time consumption for similarity calculation is not shown. It seems the proposed method does not need training, thus the title of Sec. 3.3 should be amended accordingly.

**Questions:**

See Weakness.

---

> ### Author Response · Authors · 2023-11-20
> **Response to Reviewer(dMbi) (1/2)**
>
> We appreciate the thorough review and suggestion of Reviewer(dMbi). We are pleased to respond to these comments and questions as below.
>
> **Q1: Motivation is not described clearly and insightfully. Why utilize fine-grained instances to estimate distributions?**
>
> A1: Thank you for your comments.
> Our motivation stems from the limitation of the well-known LRDC approach [Yang et al., 2021]. LRDC utilizes similar categories to estimate the distribution of the support samples. However, as we illustrated in Fig. 1 of our manuscript, we have observed that these similar categories are widely separated in the feature space, besides, we have also found that the similar categories are not sufficiently close to the support sample compared to the similar instance, this would result in the distribution of similar categories accurately representing for the distribution of support samples.
> Based on these observations, we have calculated the statistics of similar categories and instances in Appendix A.1 (Tab. 4 and 5). The results show that similar categories exhibit high diversity but low similarity, while similar instances exhibit low diversity but high similarity. These results validate the unique contribution of similar instances in comparison to similar categories and also lead us to naturally question whether utilizing similar instances can estimate the distribution well and whether there are potential benefits in combining the use of similar categories and similar instances in the estimation. Toward this, we investigate the effectiveness of the mentioned similar instances and provide two positive answers for the proposed question in our ablation study as shown in Tab. 1:
> (1) Our instance-level estimation outperforms category-level estimation by 1.51\%/0.81\% in 1/5 shot settings.
> (2) The fusion of categories estimation and instance estimation achieves the best performance both in 1/5 shot setting.
>  We will provide a more detailed description and explanation of our motivation in the revised manuscript.
>
> **Q2: Utilizing HGDE to estimate distributions is simple, and increases the amount of computation but the gains are limited compared with counterparts. Also, the category estimation has been explored in the community, and this degrades the contribution and novelty of this paper.**
>
> A2: Thank you for your feedback. **a)** As you mentioned, we have proposed a simple but effective method for few-shot image classification.  It is true that utilizing HGDE to estimate distributions may introduce some additional computational overhead, and we have compared the training time consumption between our proposed HGDE and the classical and simple LRDC in the table below.
>
> ||$K$ = 1 (600 tasks) |$K$=5 (600 tasks)|
> |:----:|:----:|:----:|
> |LRDC|276.3s (68.57 $\pm$ 0.55)|412.6s (82.88 $\pm$ 0.42)|
> |**HGDE**|366.6s (69.85 $\pm$ 0.80)|368.6s (84.63 $\pm$ 0.51)|
>
> It can be observed that in the 1-shot scenario, we introduce a limited additional training time with 1.28\% accuracy improvement, while in the 5-shot scenario, our method requires less training time also with 1.75\% accuracy gains as we only conduct distribution estimation for each support prototype, while LRDC conducts distribution estimation for each support sample. This result demonstrates the efficiency of our HGDE.
>
> **b)** About the contribution and novelty of our paper, we would like to explain that our main contribution lies in addressing the limitations of category estimation in few-shot learning. Specifically, our HGDE is the first approach that explores the effectiveness of similar instances and incorporates it with similar categories in the few-shot image classification task. Besides, the category estimation in our HGDE is different from that in LRDC, because our HDGE considers the estimation noise in covariance and employs eigenvalues decomposition to eliminate the noise component as described in Sec. 3.2.3. We will ensure to clarify the significance of our approach and its contributions more explicitly in the revised manuscript.

---

> ### Author Response · Authors · 2023-11-20
> **Response to Reviewer(dMbi) (2/2)**
>
> **Q3: In section 3.2.2, the top k most similar base samples are selected based on cosine distance, if the selected samples belong to the same class as the support prototypes, is it still reasonable for learning of the distribution, or even counterproductive?**
>
>  A3: **a)** We would like to kindly emphasize that in the few-shot learning setting, the category of the base set and the support set are kept disjoint, and our HGDE method exclusively chooses samples and categories from the base set. As a result, it is expected that the selected base samples will not be from the same class as the support set.
>
>  **b)** Besides, we imply the reviewer may be expressing that the selected base samples belong to the same class as the selected category prototypes. This would occur indeed as we illustrate in Fig. 1. Moreover, the selected samples exhibit higher similarity with the novel sample compared to its corresponding category prototypes, so it would not lead to negative impacts. The results in Tab. 1 further indicate that the instance-level estimation surpasses the category-level estimation in both $K$=1,5 settings.
>
>  **c)** Furthermore, following the suggestions, we have conducted additional experiments. As indicated in the table below, selecting samples that belong to the same category of the support prototypes does not result in a performance decline; instead, it produces superior outcomes.
>
> ||$K$ = 1|$K$ = 5|
> |:----:|:----:|:----:|
> |**HGDE**(samples belong to base set)|69.85 $\pm$ 0.80|84.63 $\pm$ 0.51|
> |**HGDE**(samples belong to the same category)|72.32 $\pm$ 0.33|86.49 $\pm$ 0.25|
>
> However, it is essential to emphasize that this strategy would entail data leakage, which is practically unfair in the few-shot learning setting.
>
>  **Q4: What are the pre-trained datasets of the chosen feature-extracting networks? Also, the increased time consumption for similarity calculation is not shown. It seems the proposed method does not need training, thus the title of Sec. 3.3 should be amended accordingly.**
>
> A4: **a)** In the setting of few-shot learning, the datasets are divided into two disjoint parts based on categories. The base set consists of base categories and is used to train the feature extraction network from scratch. The remaining data is used to train the few-shot classifier and for testing. Therefore, in the context of classical few-shot learning, there is no scenario where a network is pre-trained on a large-scale dataset and then transferred to other few-shot datasets.
> We additionally list the details of datasets used in our work, as well as its split manner as below  (also stated in Section 4.1):
> MiniImagnet： comprising 100 categories，64 categories are regarded as base datasets (categories) and used for training the feature-extracting networks.
> TieredImagenet： containing 608 categories, 351 categories are regarded as base datasets (categories) and used for training the feature-extracting networks.
> CUB： including 200 categories, 100 categories are split as base datasets (categories) and used for training the feature-extracting networks.
> CIFAR-FS： comprising 100 categories, 64 categories are split as base datasets (categories) and used for training the feature-extract networks.
>
> **b)** As for the time consumption, we have calculated time spent on category (distribution) estimation, and instance (distribution) estimation, respectively, and we have also compared the time consumption between LRDC and our HGDE (with and without similarity calculation). The results are shown below:
>
> ||$K$ = 1 (600 tasks) |$K$=5 (600 tasks)|
> |:----:|:----:|:----:|
> |Category Estimation|272.2s|273.6s|
> |Instance Estimation|303.7s|305.8s|
> |LRDC|276.3s|412.6s|
> |**HGDE**(without similarity calculation)|361.3s|363.7s|
> |**HGDE**(overall calculations)|366.6s|368.6s|
>
> It can be observed that our instance estimation requires slightly more time than category estimation in both the 1-shot and 5-shot scenarios. Besides, the time consumption of similarity calculation can be disregarded in the overall calculation in HGDE.
> In the comparison of LRDC and HGDE, our HGDE introduces a limited additional training time in 1-shot, while in the 5-shot scenario, HGDE requires less training time, demonstrating its efficiency. We will provide a more detailed analysis of the time consumption in the revised manuscript.
>
> **c)** We appreciate the comments regarding the title of Section 3.3. Our proposed HGDE is a training-free method and only includes matrix multiplication for category-level and instance-level distribution estimation. Besides, the overall pipeline can include a trainable classifier, i.e. a fully connected layer or a Logistic Regression classifier. We will make sure to revise the statement in Section 3.3 to prevent any misunderstanding.

---

> > ### Author Response · Authors · 2023-11-23
> > **Addressing concerns**
> >
> > Dear reviewer dMbi,
> >
> > Thank you again for your thorough review of our submission. As we approach the end of the author-reviewer discussion phase, we would be grateful if you could inform us whether the responses and revisions paper have addressed your concerns. We would be more than happy to provide further clarifications and revisions if you have any more questions or concerns.

---

### Comment · Area_Chair_VuVq · 2023-11-22
**Let's have (more) discussion with authors**

Dear reviewers,

The author-reviewer discussion period is closing at the end of Wednesday Nov 22nd (AOE). Let's take this remaining time to have (more) discussions with the authors on their responses to your reviews. Should you have any further opinions, comments or questions, please let the authors know asap and this will allow the authors to address them.

Kind regards, AC

---

### Author Response · Authors · 2023-11-22
**Summary of Revisions**

We really appreciate all reviewers for their helpful feedback and suggestions. We have responded to their comments individually.
We have uploaded a revised manuscript incorporating valuable feedback from reviewers. Below is a summary of the main changes:

* The experiments on different selection strategies and additional baselines are provided in Table 7 (in Appendix). (Reviewer dMbi & Reviewer Gd5o)
* The experiments on time consumption are included in Table 8 (in Appendix). (Reviewer dMbi, Reviewer Gd5o, & Reviewer 3X7E
)
* We have revised the description of our motivation and explained the reasonability of using instance-level estimation. (Reviewer dMbi)
* The experiments on applying HGDE to the prompt-based methods are included in Table 9 (in Appendix). ( Reviewer Gd5o)
* The experiments on metric results between generated samples by our HGDE and the ground truth are provided in Table 2. ( Reviewer Gd5o)
* We have revised the title of  Section 3.3 and provided the description of Figure 1. (Reviewer dMbi &Reviewer 3X7E)

We really hope our responses and revisions address all reviewers’ concerns!  In any case, we remain available to answer any further questions.

---

### Meta-Review · Area_Chair_VuVq · 2023-12-08

**Metareview:**

Based on the submission, reviews, and author feedback, the main points raised for this work are summarised as follows.

Strengths:

1. The paper is easy to follow and the illustrations are effectively utilised.
2. The ablation study is enough and easy to follow.
3. The proposed HGDE has the significant advantage of integrating the statistics from both category- and instance-level data.
4. The proposed refinement process helps to achieve more precise estimates of mean and covariance.

Issues:

1. The contribution and novelty of this work are not significant enough.
2. The concerns related to the basic hypothesis, the gain over SMKT, and the large variance.
3. The proposed method appears to be more complex than some existing methods.

The authors provided feedback and it helps to address some of the raised issues. After reading this submission, I feel that overall this is a solid piece of work, with clear motivation and technically sound solutions. The ablation study is extensive in showing the properties of the proposed method. The experimental comparison shows the improved performance. Meanwhile, the proposed method is relatively straightforward and its technical contribution, although good, does not seem to be significant enough. In addition, some details of the proposed method could be further examined.

Regards, AC

**Justification For Why Not Higher Score:**

1. The ratings of this submission are not high (5, 5, 6). Although one reviewer gives 6, no one argues for this work during discussion.
2. The proposed method is relatively straightforward and its technical contribution does not seem to be significant enough.
3. Some details of the proposed method could be further examined.

**Justification For Why Not Lower Score:**

N/A

---

### Decision · Program_Chairs · 2024-01-16

Reject